# Effects of Freshwater Acidification on the Gut Microbial Community of *Trachemys scripta elegans*

**DOI:** 10.3390/ani14131898

**Published:** 2024-06-27

**Authors:** Xin Niu, Zhaohui Dang, Meiling Hong, Haitao Shi, Li Ding

**Affiliations:** Ministry of Education Key Laboratory for Ecology of Tropical Islands, Key Laboratory of Tropical Animal and Plant Ecology of Hainan Province, College of Life Sciences, Hainan Normal University, Haikou 571158, China; 202311071300001@hainnu.edu.cn (X.N.); 15834308829@163.com (Z.D.); haitao-shi@263.net (H.S.)

**Keywords:** freshwater acidification, gut microbiota, *Trachemys scripta elegans*, gut microbiome health index (GMHI), microbial dysbiosis index (MDI)

## Abstract

**Simple Summary:**

Using high-throughput Illumina sequencing, this research explored how freshwater acidification affects the gut microbiota of *Trachemys scripta elegans*. It finds that shifts in pH cause notable changes in the microbes’ composition and their ability to handle stress and disease. The study emphasizes the pH sensitivity of microbial communities and how acidification might upset this balance, impacting aquatic species’ health. This understanding is key for managing freshwater ecosystems’ ecological effects of acidification and for conservation efforts to protect their biodiversity.

**Abstract:**

Freshwater acidification (FA) has become a global environmental problem, posing a potential threat to freshwater ecosystems. The gut microbiota plays a crucial role in the host’s response and adaptation to new environments. In this study, we investigated the changes in microbial communities in Red-eared slider (*Trachemys scripta elegans*) under acidic conditions to reveal the ecological impacts of acidification on freshwater turtles. The results showed that there were significant differences in β-diversity (*p* = 0.03), while there were no significant differences in the α-diversity of gut microbiota in *T. s. elegans* between the different levels of acidification (pH of 5.5, 6.5, 7.5). Both the Gut Microbiome Health Index (GMHI) and the Microbial Dysbiosis Index (MDI) exhibited significant differences when comparing environments with a pH of 5.5 to those with a pH of 6.5 (*p* < 0.01). A comparative analysis between pH levels of 5.5 and 6.5 also revealed substantial differences (*p* < 0.01). Likewise, a comparative analysis between pH levels of 6.5 and 7.5 also revealed substantial differences (*p* < 0.01). At the phylum level, Firmicutes, Fusobacteria, and Bacteroidota formed a major part of the gut microbial community, Fusobacteria showed significant differences in different acidity environments (*p* = 0.03). At the genus level, *Cetobacterium*, *Turicibacter*, unclassified Eubacteriaceae, and *Anaerorhabdus*_*furcosa*_group showed significant differences in different acidity environments. The pH reduced interactivity in the gut microbiota of *T. s. elegans*. In addition, LEfSe analysis and functional prediction revealed that the potentially_pathogenic and stress_tolerant functional characteristics also showed significant differences in different acidity environments. The findings underscore the pivotal role of the gut microbiota in *T. s. elegans* in response to freshwater acidification and provide a foundation for further exploration into the impacts of acidification on freshwater ecosystems.

## 1. Introduction

Anthropogenic-induced increases in atmospheric CO_2_ concentrations have been recognized as the primary driver of ocean acidification [1] and are similarly considered to be a key factor contributing to the acidification of freshwater ecosystems [2,3]. The impacts of ocean acidification on marine ecosystems have been extensively studied, including significant effects on food web structure, nutrient cycling, productivity, and biodiversity [4,5]. However, despite significant progress in ocean acidification research, understanding of the response and adaptation mechanisms of freshwater ecosystems to anthropogenic-induced CO_2_ emissions remains relatively limited. Existing studies have shown that freshwater acidification (FA) has a significant impact on zooplankton community structures in freshwater ecosystems [6] and on inducible defense mechanisms in key species such as Daphnia [2]. In addition, the sensitivity of freshwater fishes to acidifying environments has been demonstrated, with acidifying conditions impairing physiological functions in juvenile pink salmon (*Oncorhynchus gorbuscha*) [2], reduced oxygen-carrying capacity of the blood being observed in rainbow trout (*Oncorhynchus mykiss*) [7], and the development of necrotic gill lesions and hepatitis being noted in carp (*Cyprinus carpio*) [8], and acidification of freshwater poses a potential risk to fish embryo viability by increasing their susceptibility to silver toxicity [9]. Nevertheless, research on the effects of FA on other freshwater organisms such as turtles is still limited. As an important group in freshwater ecosystems, the sensitivity and adaptation of turtles to the environmental changes in FA also deserve attention.

Turtles and tortoises are abundant worldwide, with some 357 species, but this diversity is facing serious challenges and it is estimated that approximately 51.3% of turtle and tortoise species are threatened with extinction (TFTSG; www.iucn-tftsg.org (accessed on 26 May 2024)). The red-eared slider (*Trachemys scripta elegans*) is distributed in various aquatic ecosystems worldwide [10] and has demonstrated a strong ability to survive and reproduce in a wide range of freshwater environments, maintaining normal physiological functions even in polluted waters [11,12,13]. This strong tolerance allows *T. s. elegans* to survive under a wide range of environmental conditions. However, *T. s. elegans* may still suffer varying degrees of negative effects when exposed to different environmental stressors. For other turtle and tortoise species, environmental threats may be more severe because they lack this tolerance in *T. s. elegans*.

Microbial communities play a crucial role in host health, as they are not only involved in regulating the host’s immune system and promoting metabolism but also assist the host in resisting various environmental stresses [14,15,16]. Microbial–host interactions are complex and finely tuned and play an integral role in maintaining the host’s state of health [17]. However, an imbalance in the microbial community can lead to a deterioration in host health and has been strongly linked to the development of a variety of diseases [18,19]. A variety of environmental factors can lead to dysbiosis in the gut microflora of animals; for example, high temperatures have been shown to cause changes in the gut microflora of animals, especially in the phylum Firmicutes and Proteobacteria [20]. In the gut tract of *T. s. elegans*, a wide range of microorganisms are present, including *Aeromonas*, *Hafnia*, *Salmonella*, *Enterobacter*, and *Clostridium perfringens* [21], of which Firmicutes, Bacteroidota, and Spirochetes form a major part of the gut microbial community [22]. Although studies have begun to focus on the effects of environmental change on the gut microbial communities of *T. s. elegans* [23,24], there is still relatively little research on how these microbial communities respond to FA.

This study explores the effects of freshwater acidification on the gut microbiota of *T. s. elegan*, particularly how it influences microbial diversity and functionality across a range of pH levels. The research seeks to delineate the ecological impacts of acidification and to elucidate the adaptive responses of the gut microbiome to environmental stress. The findings are instrumental for formulating conservation strategies aimed at safeguarding aquatic biodiversity and ecosystem services, underscoring the critical role of gut health amidst environmental perturbations.

## 2. Materials and Methods

### 2.1. Research Animals

The experimental animals *T. s. elegans* were purchased from Hongwang Turtle Farm in Dongshan Town, Haikou, and domesticated in a breeding room for four weeks. The water temperature in the breeding room was 26–28 °C and the dissolved oxygen (DO) was not less than 5 mg/L. LED lights were installed in the hatchery to simulate natural light, providing 14 h of light and 10 h of darkness per day. *T. s. elegans* with a weight range of 150 g ± 30 g (29 turtles) were selected and randomly divided into three groups and fed in different acidic environments: control group (pH = 7.5 ± 0.2, 10 turtles), experimental group one (pH = 6.5 ± 0.2, 10 turtles), and experimental group two (pH = 5.5 ± 0.2, 9 turtles). Both the control and experimental groups were reared in a 1 m × 0.5 m turtle rearing tank in the turtle rearing room of Hainan Normal University. During the experiment, the carbon dioxide vacuum pump was connected to a bubble stone, and the size was controlled by a valve. Carbon dioxide was continuously injected to maintain a stable pH in the tank, and the pH of the tank was checked three times a day to control the pH required for the experiment. All three groups were fed the same regular diet throughout the exposure period. With the approval of the Animal Ethics Committee of the Hainan Ecological Environment Education Centre (2023-005), after 60 days of acidification treatment, all *T. s. elegans* were fasted for 3 days beforehand. They were collected from each group and anesthetized with pentobarbital sodium. The gut contents were frozen in 2 mL EP tubes, snap-frozen in liquid nitrogen, and then transferred to a −80 °C refrigerator for later use. The 16S rRNA gene detection was performed on all samples.

### 2.2. DNA Extraction and PCR Amplification

Microbial genomic DNA was extracted from gut content samples using a MagAtractPowerSoil Pro DNA Kit (Qiagen, Hilden, Germany) according to the manufacturer’s instructions. The quality and concentration of DNA were determined using 1.0% agarose gel electrophoresis and a NanoDrop2000 spectrophotometer (NanoDrop2000, Thermo Scientific, Boston, MA, USA), and it was stored at −80 °C until subsequent utilization. The V3–V4 highly variable portion of the bacterial 16S rRNA gene was amplified using an ABI GeneAmp^®^ 9700 PCR Thermal Cycler (GeneAmp 9700, ABI, Carlsbad, CA, USA). For bacterial amplification, the primers 338F (5′-ACTCCTACGGGGAGGCAGCAG-3′) and 806R (5′-GGACTACHVGGGTWTCTAAT-3′) were employed [25]. PCR reactions were conducted three times per sample in 20 µL reactions. The PCR reaction mixture comprised Taq Pro Multiplex DNA Polymerase (10 µL), template DNA (10 ng), and each primer (5 µM, 0.8 µL), adjusted to 20 µL with ddH_2_O. PCR amplification cycling conditions were an initial denaturation at 95℃ for 3 min, followed by 29 cycles. The denaturation was conducted at 95 °C for 30 s, followed by annealing at 53 °C, extension at 72 °C for 30 s, and a final single extension at 72 °C for 10 min, concluding at 4 °C. Subsequently, the PCR product was extracted from a 2% agarose gel, purified using the PCR Clean-Up Kit (YuHua, Shanghai, China) following the manufacturer’s instructions, and quantified using a Qubit 4.0 (Qubit 4.0, Thermo Fisher Scientific, Waltham, MA, USA). The purified amplicons were pooled in equimolar quantities. Subsequently, they were sequenced using a paired-end approach on an Illumina PE300 platform (Illumina, San Diego, CA, USA) following the standard protocols of Majorbio Bio-Pharm Technology Co., Ltd. (Shanghai, China). The raw sequencing reads were deposited in the NCBI Sequence Read Archive database (Accession Number: SRR28441908).

### 2.3. Processing of Sequencing Data

The raw sequences underwent fastp software quality control [26]. They were spliced using FLASH software (1.2.11) [27]. To enhance read quality, we filtered bases with a quality value of ≤20 from the ends of the reads. We implemented a 50 bp window and trimmed bases from the back end if the average quality value within the window was <20. Additionally, we excluded reads shorter than 50 bp and those containing N-bases after quality control. Paired-end reads were merged into a single sequence based on the overlapping relationship between the reads, with a minimum overlap length of 10 bp. The maximum mismatch ratio of 0.2 was allowed in the overlapping region of merged sequences, and non-conforming sequences were filtered out. Samples were distinguished based on the barcode and the primer sequences at the first and last end, adjusting the sequence orientation. The barcode allowed zero mismatches, while up to two primer mismatches were permitted. Operational Taxonomic Units (OTUs) were clustered at a 97% similarity threshold, and chimeras were removed using UPARSE v7.1 software [28,29]. More information about UPARSE (11) can be found at http://drive5.com/uparse/ (accessed on 22 April 2024). Sequences annotated to the chloroplast and mitochondrial sequences were removed from all samples. Taxonomic annotation of OTU was conducted using the Silva 16S rRNA (v138) and ITS (Unite v.8.0) gene databases. The RDP classifier [30] (http://rdp.cme.msu.edu/ (accessed on 22 April 2024), version 2.13) was employed with a confidence threshold of 70%. The community composition of each sample was then determined at various taxonomic levels.

### 2.4. Ecological and Statistical Analysis

We utilized Mothur (v.1.30.2, University of Michigan, Ann Arbor, MI, USA) to generate sparse curves, assessing the sequencing depth adequacy to cover the estimated number of OTUs at 97% sequence similarity [31]. Subsequently, the data were analyzed using Mothur for α-diversity indices, evaluating bacterial and fungal community abundance and diversity. Kruskal–Wallis rank-sum tests were used to assess differences in α-diversity. The non-metric multidimensional scaling sorting (NMDS) method was applied to assess β-diversity. To assess the likelihood of disease in *T. s. elegans* across varying pH levels, this study employs the Gut Microbiome Health Index (GMHI), a biologically interpretable mathematical model that predicts disease susceptibility irrespective of clinical diagnosis, as described by Gupta et al. in 2020 [32]. Concurrently, to evaluate the extent of microbial dysbiosis under these pH conditions, the Microbial Dysbiosis Index (MDI) is calculated, following the methodology established by Gevers et al. in 2014 [33]. OTU Venn diagrams were generated using the BASE and VEGAN packages in R [34]. Analysis of community structure at the phylum and genus levels was conducted using the PANDAS package in Python (v.2.7). We utilized the data table from the tax_summary_a folder for analysis. A bar chart was generated, merging relative abundances < 1% into the category ‘others’ [34]. Significant differences between the gut microbes of *T. s. elegans* in different pH groups were calculated using one-way ANOVA, and *p* < 0.05 was considered statistically significant. Linear discriminant analysis (LDA) was conducted on samples grouped under different conditions, with a significance level set at one-way ANOVA *p*-value < 0.05 and an LDA score > 2 [35]. Based on the relative abundance of species OTUs, the Networkx complex network analysis toolkit was used to construct a single-factor correlation network, and then Gephi software (0.9.2) was used to construct a cooccurrence network map. Next, the SparCC algorithm from SpiecEasi [36] was applied to perform correlation analysis on the top 100 genera based on relative abundance. This involved conducting 20 iterations to compute the sparse correlations among genera and estimating pseudo *p*-values through 100 bootstraps. Correlation coefficients and *p*-values were separately computed for microbial community members in the different pH groups. Finally, a network was constructed by selecting edges with |correlation| > 0.5 and *p* < 0.05 (two-sided), followed by visualization using Cytoscape software (V3.9.0). We employed PICRUSt2 to predict functional differences in the gut bacterial communities of *T. s. elegans* in different pH groups, leveraging 16S rRNA gene sequences to predict KEGG immediate homolog (KO) functional profiles [37]. Microbial communities were analyzed using BugBase [38], a microbiome tool identifying phenotypic levels and predicting microbial traits. BugBase normalizes OTUs based on the number of predicted 16S copies and uses a precalculated file for predictions. Relative abundance shifts of *T. s. elegans* gut microbes across pH groups were compared using the one-way ANOVA, with significance set at *p* < 0.05.

## 3. Results

### 3.1. Analysis of 16S rRNA Sequencing Results

The assay yielded 41451–64441 quality-filtered sequences obtained per sample, totaling 1,379,785 sequences (440,337 reads in pH 5.5 group, 469,503 reads in pH 6.5 group, and 469,945 reads in pH 7.5 group). Sequences had a length distribution of 405–417 bp, with an average of 410 bp (Appendix A). All samples were sparse to 33,079 reads sequencing (Appendix A). All sparse curves reached the saturation stage, indicating that an adequate sampling depth was achieved for each sample.

### 3.2. Alpha and Beta Diversity Analyses

Figure 1 shows the α-diversity indices (Shannon, Chao, Simpson and Ace) in *T. s. elegans* in different pH groups. There were no significant differences in Shannon, Chao, Simpson and Ace between different groups (*p* > 0.05; Figure 1).

β-diversity analysis was performed using the NMDS analysis plot based on weighted UniFrac distances. The gut flora of *T. s. elegans* overlapped but also differed across pH groups, and the differences were significant (*p* = 0.03, Figure 2).

### 3.3. The Gut Microbiome Health Index and Microbial Dysbiosis Index Analysis

In *T. s. elegans,* GMHI shows significant differences between pH 5.5 and pH 7.5 (*p* < 0.001; Figure 3A). Similarly, there are significant differences in the GMHI between pH 6.5 and pH 7.5 (*p* < 0.001; Figure 3B). Additionally, the MDI for *T. s. elegans* also exhibits significant differences between pH 5.5 and pH 7.5 (*p* < 0.01; Figure 3C). Again, significant differences are observed in MDI between pH 6.5 and pH 7.5 (*p* < 0.01; Figure 3D).

### 3.4. Colony Composition and Relative Abundance

Venn diagrams illustrate the shared and distinct gut flora of *T. s. elegans* in different pH groups. In the gut flora, 741 OTUs were found, with 417 (56.28%) shared in different pH groups (Figure 4). Among these shared OTUs, 273 belonged to Firmicutes (65.47%), 34 to Bacteroidota (8.15%), and 31 to Actinobacteriota (7.43%). In the pH 5.5 group, 36 OTUs were identified as unique, comprising 4.86% of the total. Among these, 15 OTUs belonged to Firmicutes (41.67% unique), 7 to Proteobacteria (19.44% unique), and 4 to Bacteroidota (11.11% unique). In the pH 6.5 group, 26 OTUs were identified as unique, comprising 3.51% of the total. Among these, 14 OTUs belonged to Firmicutes (53.85% unique), 4 to Proteobacteria (15.38% unique), and 2 to Bacteroidota (7.69% unique). In the pH 7.5 group, 101 OTUs were identified as unique, comprising 13.63% of the total. Among these, 27 OTUs belonged to Bacteroidota (26.73% unique), 20 to Firmicutes (19.80% unique), and 20 to Proteobacteria (19.80% unique).

The dominant phylum (>1%) of the gut flora in *T. s. elegans* in different pH groups were Firmicutes, Fusobacteriota, and Bacteroidota (Figure 5A). Significant differences were observed in Fusobacteriota in *T. s. elegans* in different pH groups (Kruskal–Wallis, *p* < 0.05) (Figure 5C).

In *T. s. elegans* in different pH groups, genera such as *Cetobacterium*, unclassified Bacteroidales, *Romboutsia*, and *Clostridium_sensu_stricto_1* exhibited a higher abundance of gut flora (Figure 5B). Significant differences in *T. s. elegans* in different pH groups were observed in the genera *Cetobacterium*, *Turicibacter*, unclassified Eubacteriaceae, and *Anaerorhabdus_furcosa*_group (Figure 5D). The relative abundance of *Cetobacterium*, unclassified Eubacteriaceae, and *Anaerorhabdus_furcosa*_group was significantly higher in the pH 7.5 group than in the pH 6.5 group and pH 5.5 group (*p* < 0.05, Figure 5D). The relative abundance of the genus *Turicibacter* was significantly higher in the pH 5.5 group than in the pH 6.5 group and pH 7.5 group (*p* < 0.05, Figure 5D).

Differences in the relative abundances of *T. s. elegans* in different pH groups were determined based on LEfSe analyses. From the phylum to the genus level, a total of 23 differential features were identified, of which 17 differential features were in the pH 7.5 group, 4 differential features were in the pH 6.5 group and 2 differential features were in the pH 5.5 group (Figure 6). In the pH 5.5 group, *T. s. elegans* gut microbes are significantly enriched in Micrococcaceae and *Arthrobacte*. In the pH 6.5 group, *T. s. elegans* gut microbes are significantly enriched in *Thermomicrobiales* and Frankiales. In the pH 7.5 group, *T. s. elegans* gut microbes are significantly enriched in Fusobacteria, Fusobacteriota, and Fusobacteriales.

### 3.5. Gut Microbiota Interaction Network Analysis

The pH 5.5 group showcased a gut microbial network in *T. s. elegans* with 50 nodes and 86 edges (Figure 7A). The pH 6.5 group showcased a gut microbial network in *T. s. elegans* with 45 nodes and 114 edges (Figure 7B). The pH 7.5 group showcased a gut microbial network in *T. s. elegans* with 50 nodes and 200 edges (Figure 7C). As the pH drops reduced interactivity in the gut microbiota of *T. s. elegans*. To further identify the key bacteria under pH, a univariate correlation network of the top 50 genera was constructed based on the correlation between species. This network is reflected in the interaction relationships among species in the samples (Figure 7D). The results show that the genus *UCG-002* had the highest connectivity and was closely linked to 10 other genera among the top 50 genera.

### 3.6. Predicted Functional Analysis

The microbiota in the guts of *T. s. elegans* in different pH groups exhibited several key functions, including metabolism, genetic information processing, environmental information processing, human diseases, cellular processes, and organismal systems (Figure 8A). *T. s. elegans* in different pH groups exhibited nonsignificant differences in global and overview maps, translation, carbohydrate, amino acid, energy, cofactor, and vitamin metabolisms (Kruskal–Wallis, *p* > 0.05) (Figure 8B). To clarify the differential changes in the gut flora of *T. s. elegans* in different pH groups, we employed the BugBase algorithm to analyze and predict bacterial phenotypes. This allowed us to explore significant differences in the functions and characteristics of the gut flora in different pH groups, focusing on traits such as Potentially_Pathogenic and Stress_Tolerant (Kruskal–Wallis, *p* < 0.05) (Figure 8C). In addition, the proportion of bacteria related to the Potentially_Pathogenic and Stress_Tolerant traits gradually increased with the pH drops, and there was a significant difference between the pH 5.5 group and pH 7.5 group (Figure 8D).

## 4. Discussion

The functional analysis of animal gut microbiota plays a crucial role in revealing the physiological functions and ecological adaptability of the host [20,39,40,41]. These microbial communities not only play a key role in the host’s nutritional metabolism but also participate in the regulation of the immune system and may influence the host’s adaptability to environmental changes [42]. In this study, we utilized high-throughput Illumina sequencing technology to delve into the impact of freshwater acidification on the gut microbial community of *T. s. elegans*. The results indicated significant changes in the composition of the gut microbiota under different acidity conditions. These alterations not only reflect the high sensitivity of the gut microbial community to environmental acidification but also have profound implications for the host’s physiological health. The insights gained from this study are essential for understanding the complex dynamics of gut microbiota in response to environmental stress and may inform future strategies for mitigating the adverse effects of acidification on aquatic species.

This study found no significant differences in α-diversity among the gut microbial communities of *T. s. elegans* under varying degrees of acidification. This is consistent with the research results of Wang’s study on *Exopalaemon carinicauda* in 2023 [43]. Similarly, Kong’s study on the Pacific oyster *Crassostrea gigas* in 2022 also reached a similar conclusion [44]. However, significant differences were observed in β-diversity, primarily due to changes in the diversity and relative abundance of the Fusobacteria. In an environment with a pH 7.5 group, the gut microbial community of *T. s. elegans* exhibited higher richness, consistent with previous findings that a decrease in environmental pH leads to a reduction in gut microbial community richness [45]. At the same time, the findings from this study reveal a pronounced impact of freshwater acidification on the gut microbiota of *T. s. elegans*, as evidenced by significant differences in GMHI across various pH levels. Notably, a comparison between the acidic condition of pH 5.5 and the near-neutral condition of pH 7.5 yielded highly significant disparities in GMHI, suggesting a substantial shift in the microbial community’s health status with increasing acidity. This trend is mirrored in the comparison between pH 6.5 and pH 7.5, where GMHI also demonstrated significant differences, indicating that even moderate changes in pH can elicit notable alterations in gut microbiome health. A study has shown that low pH values can lead to a decrease in the number of probiotics in the intestinal tract, thereby promoting the proliferation of pathogenic microorganisms, which may ultimately disrupt the physiological health of oysters [44]. Moreover, MDI, which quantifies the degree of microbial imbalance, followed a similar pattern, showing significant differences between pH 5.5 and pH 7.5 and again between pH 6.5 and pH 7.5. Similarly, the research conducted by Fonseca in 2019 is consistent with the finding that low PH values can lead to intestinal flora imbalance in seabreams (*Sparus aurata*) [46]. These results underscore the sensitivity of the gut microbiota to environmental pH and its potential role as a bioindicator of ecosystem health. The observed dysbiosis could have cascading effects on the host’s immune function and overall health, highlighting the need for further research into the adaptive mechanisms of gut microbiota in response to acidification and the development of targeted interventions to support the resilience of aquatic species in acidic environments.

The genus *Cetobacterium*, an important symbiotic bacterium, can produce short-chain fatty acids (SCFAs) and vitamin B12, which have a positive effect on the intestinal health and overall metabolism of the host animal [47,48]. Additionally, *Cetobacterium* can enhance the host’s resistance to viral infections [49]. In this study, the abundance of *Cetobacterium* in the gut of *T. s. elegans* varied significantly under different acidity conditions and decreased with a reduction in pH value, suggesting that an acidic environment may lower the immune capacity of *T. s. elegans* and increase its susceptibility to viral infections. Qi (2023) reached a similar conclusion with zebrafish [50]. The genus *Turicibacter* is associated with the development of inflammatory bowel disease (IBD) and may exacerbate the pathological symptoms of IBD by promoting inflammatory processes and compromising the integrity of the intestinal barrier [51]. This study found that the abundance of *Turicibacter* in the gut of *T. s. elegans* varied significantly under different acidity conditions and increased with a decrease in pH value, implying that an acidic environment may increase the risk of IBD in *T. s. elegans*. Bacteria of the family Eubacteriaceae play an important role in nutritional metabolism and maintaining intestinal balance, capable of fermenting dietary fiber to produce short-chain fatty acids with anti-inflammatory effects, particularly butyrate, which helps reduce the risk of inflammatory diseases [52]. Overall, bacteria of the Eubacteriaceae family contribute to maintaining overall gut health and preventing various diseases [53]. In this study, the abundance of undefined Eubacteriaceae bacteria in the gut of *T. s. elegans* varied significantly under different acidity conditions and decreased with a reduction in pH value, indicating that an acidic environment may disrupt the gut health of *T. s. elegans* and increase its risk of disease. Similarly, studies have shown that when compared with healthy dogs, intestinal lymphoma subjects showed significant increases in organisms belonging to the Eubacteriaceae family [54]. The *Anaerorhabdus furcosa* group within the phylum Bacteroidota occupies an important position in the gut microbiome of animals and plays a key role in the health and disease status of the host [55]. This bacterial group can produce large amounts of acetic acid, a short-chain fatty acid that inhibits the growth of certain pathogens [55]. In this study, the abundance of the *Anaerorhabdus furcosa* group in the gut of *T. s. elegans* varied significantly under different acidity conditions and decreased with a reduction in pH value, further confirming that an acidic environment may increase the risk of disease in *T. s. elegans*.

Previous studies have shown that *UCG-002* is a member of Ruminococcaceae, which has been linked to immune thrombocytopenia [56]. Under different pH groups, *UCG-002* acts as a key node in the series of bacteria. In this study, pH disrupted the equilibrium of the gut microbiota in *T. s. elegans*, manifested by a marked reduction in inter-microbiota interactions. The decreased proportion of inter-competition among gut microbiota may render *T. s. elegans* more susceptible to disease risks.

The impact of the gut microbiome on the host’s physiological functions is multifaceted, with metabolic functions being one of the most renowned and extensively studied mechanisms [57]. In this study, we observed that the gut microbial community of *T. s. elegans* exhibited a diverse array of functional characteristics under different acidity conditions, including metabolism, genetic information processing, environmental information processing, human diseases, cellular processes, and organismal systems. These findings further underscore the pivotal role of gut microbiota in the metabolic activities of *T. s. elegans*, echoing previous research outcomes on the gut microbiome of turtles [58]. Additionally, the study revealed significant differences in the potential pathogenicity and stress tolerance of *T. s. elegans* gut microbiota across varying acidity environments, with these differences becoming more pronounced as the pH value decreased. This result suggests that, in response to a decline in environmental pH, the gut microbial community of *T. s. elegans* may undergo adaptive changes that enhance its tolerance to acidic conditions. However, such adaptive shifts may come at the expense of the host’s resistance to diseases, thereby increasing the potential risk of disease in the host.

## 5. Conclusions

This study employs high-throughput Illumina sequencing technology to comprehensively explore the impact of freshwater acidification on the gut microbial community of *T. s. elegans*. Our analysis reveals significant changes in the composition and functionality of *T. s. elegans* gut microbiota under different acidity conditions, particularly in terms of potential pathogenicity and stress tolerance. These findings underscore the intricate relationship between environmental pH and microbial ecology, suggesting that even subtle shifts in acidity can precipitate substantial alterations in microbial communities. The results also imply that acidification may have broader ecological implications, potentially affecting the health and resilience of aquatic species like *T. s. elegans* by disrupting the delicate balance of their gut microbiota. This insight could inform future conservation efforts aimed at preserving freshwater ecosystems and the biodiversity they support.

## Figures and Tables

**Figure 1 animals-14-01898-f001:**
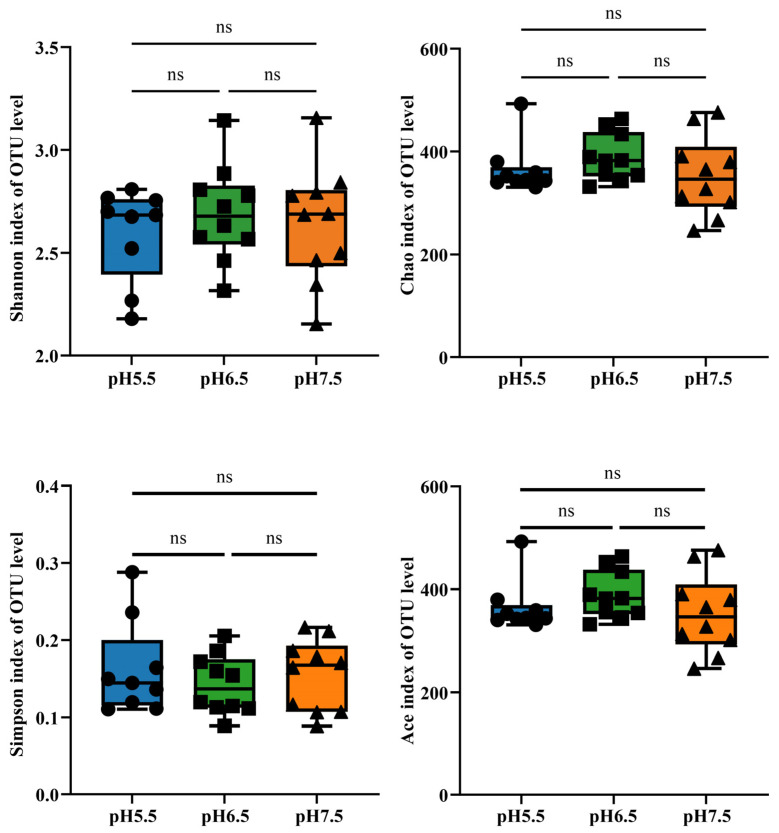
α-diversity of *T. s. elegans* in different pH groups. α-diversity by Shannon, Chao, Simpson, and Ace was tested for significance using the Kruskal–Wallis rank-sum test. *p*-values indicate the confidence level of statistical analyses, with *p* < 0.05 indicating significant differences. ns indicates nonsignificant differences.

**Figure 2 animals-14-01898-f002:**
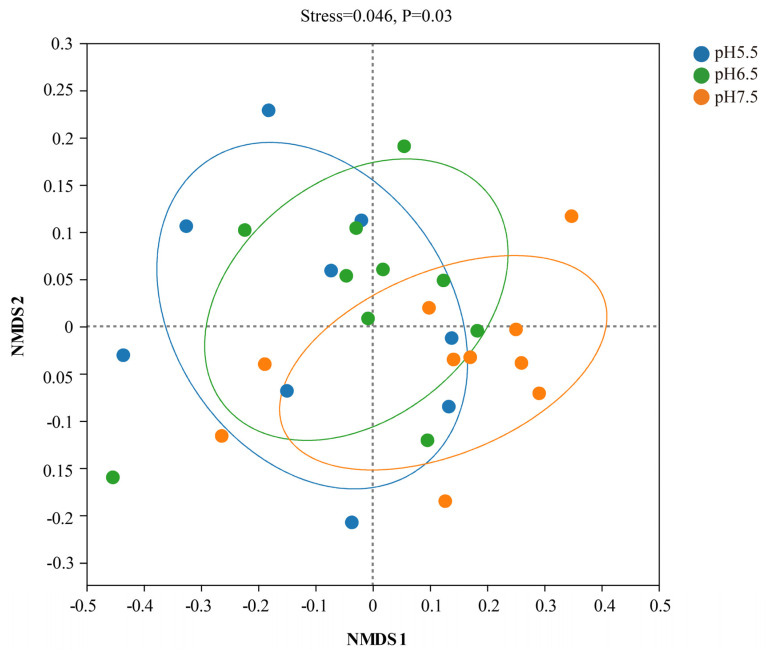
β-diversity of gut flora. NMDS analysis plot of β-diversity based on weighted UniFrac distances of *T. s. elegans* in different pH groups. It is usually assumed that stress < 0.2 can be represented by a two-dimensional dot plot of the NMDS, which is graphically interpretable; when stress < 0.1, it can be considered a good ordering and when stress < 0.05, it is well represented. *p*-values indicate the confidence level for statistical analyses, with *p* < 0.05 indicating a significant difference. Colony composition and relative abundance.

**Figure 3 animals-14-01898-f003:**
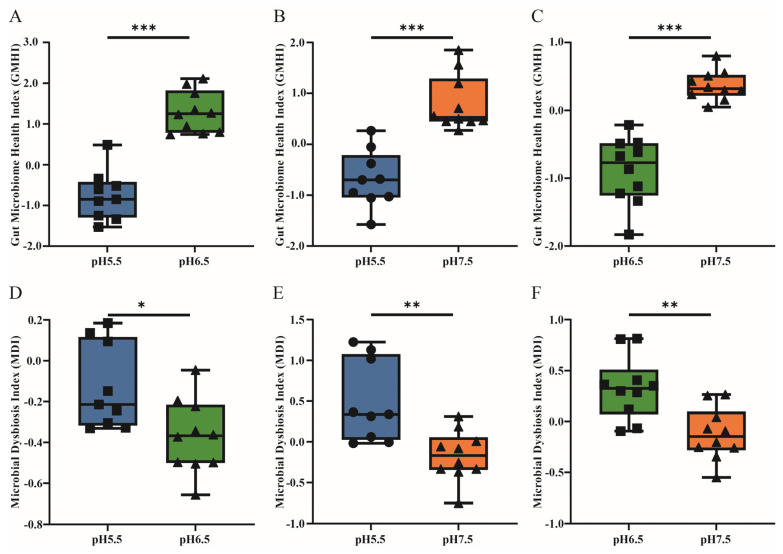
Comparative analysis of GMHI and MDI in *T. s. elegans* across different pH levels. (**A**) GMHI comparison between *T. s. elegans* exposed to pH 5.5 and pH 6.5 environments. (**B**) GMHI comparison between *T. s. elegans* exposed to pH 5.5 and pH 7.5 environments. (**C**) GMHI comparison between *T. s. elegans* exposed to pH 6.5 and pH 7.5 environments. (**D**) MDI comparison between *T. s. elegans* exposed to pH 5.5 and pH 6.5 environments. (**E**) MDI comparison between *T. s. elegans* exposed to pH 5.5 and pH 7.5 environments. (**F**) MDI comparison between *T. s. elegans* exposed to pH 6.5 and pH 7.5 environments. *p*-values indicate the confidence level of statistical analyses, with *p* < 0.05 indicating statistically significant differences. * represents *p* < 0.05, ** represents *p* < 0.01 and *** represents *p* < 0.001.

**Figure 4 animals-14-01898-f004:**
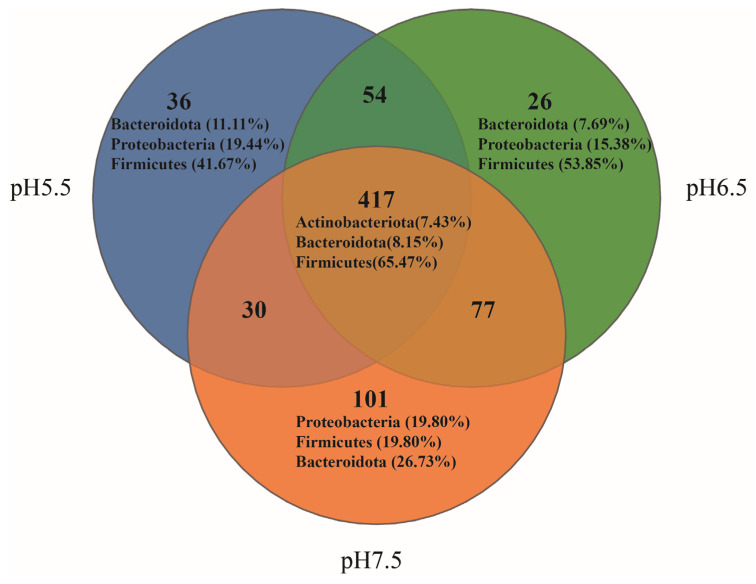
Venn diagram illustrating the unique and shared gut flora of *T. s. elegans* OTUs in different pH groups. The numbers represent the number of OTUs.

**Figure 5 animals-14-01898-f005:**
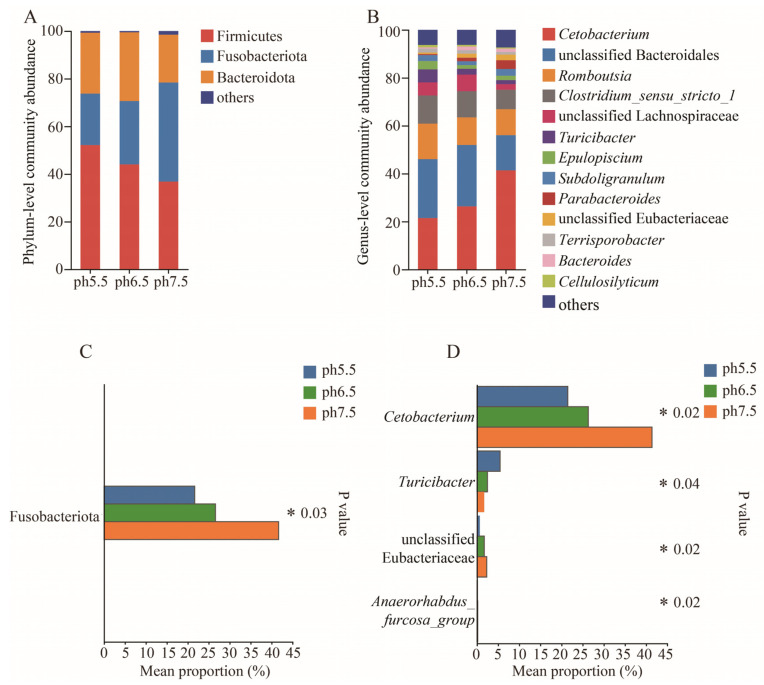
A comparative analysis of the gut flora of *T. s. elegans* in different pH groups at the phylum and genus level is presented. (**A**) illustrates the gut flora at the phylum level. (**B**) illustrates the gut flora at the genus level. (**C**) highlights significant differences in gut flora at the phylum level. (**D**) highlights significant differences in gut flora at the genus level. * represents *p* < 0.05.

**Figure 6 animals-14-01898-f006:**
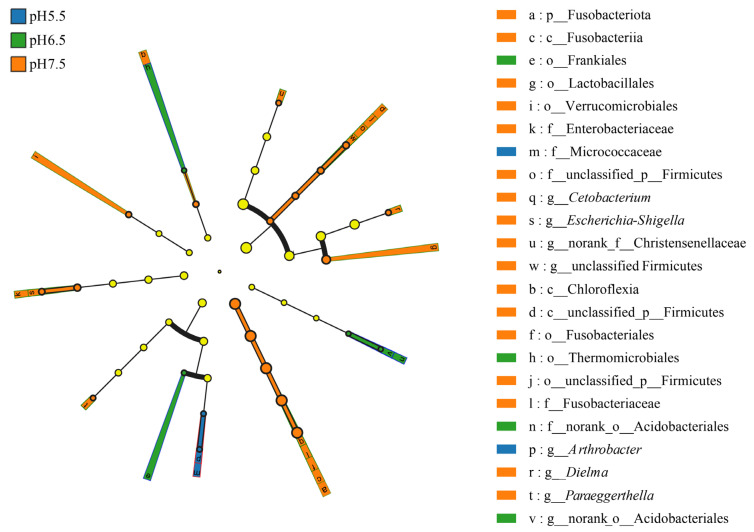
LEfSe analysis of gut flora markers in *T. s. elegans* in different pH groups. The diameter of each circle reflects its abundance. Multiclass analysis was flexible (at least one class difference). Inside-out circles reflect classification from phylum to genus. Inside-out circles symbolize the taxonomy from phylum to genus. Class, order, and family labels are displayed as well as all taxa with LDA scores > 2. The yellow color circles indicate no significant differences between the groups.

**Figure 7 animals-14-01898-f007:**
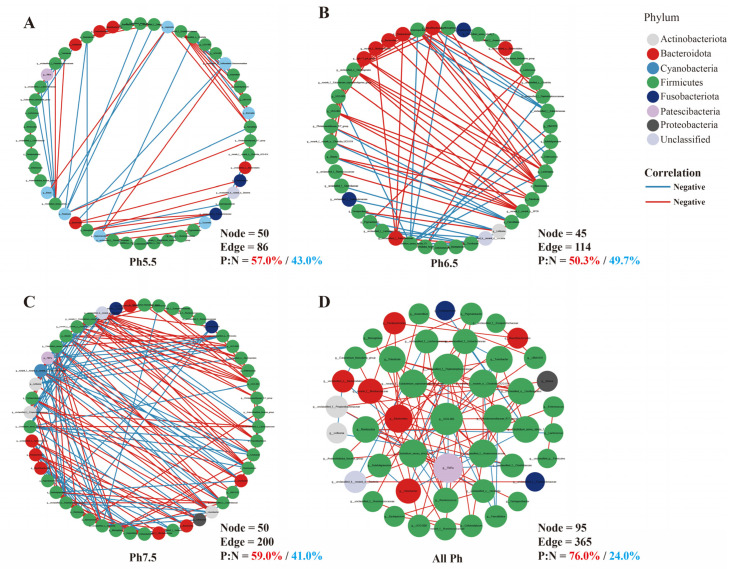
Gut microbiota genus-level interaction network under pH. (**A**) pH 5.5 group; (**B**) pH 6.5 group; (**C**) pH 7.5 group; (**D**) all pH groups. Each node represents a bacterial genus. Node colors represent bacterial phylum. Blue edges represent negative correlations, while red edges represent positive correlations (|correlation| > 0.5, *p* < 0.05).

**Figure 8 animals-14-01898-f008:**
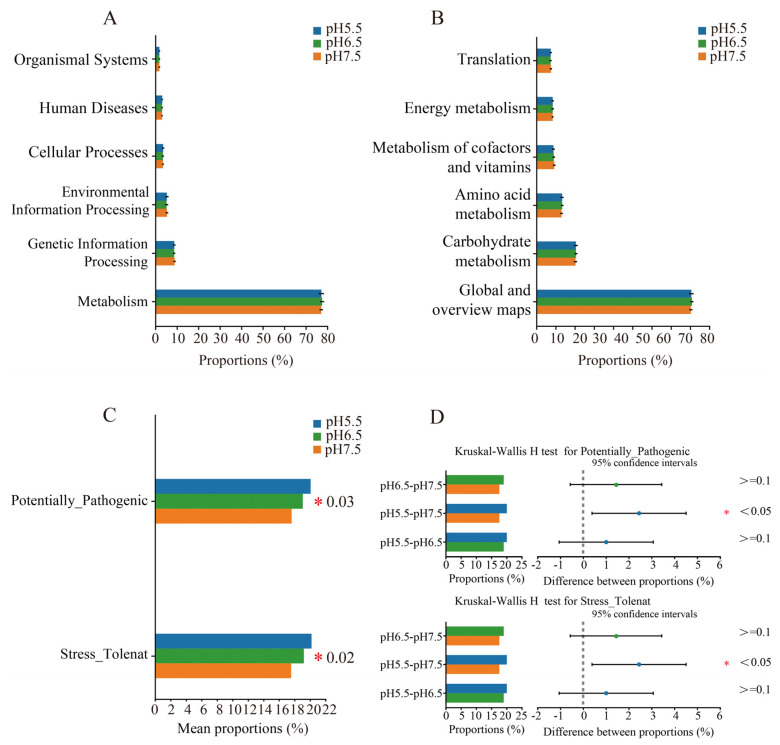
BugBase phenotype prediction of gut microbiota in *T. s. elegans* in different pH groups. Functional analysis of bacteria was conducted using PICRUSt2. Bacterial phenotypes were identified using the BugBase method. (**A**) The relative abundance of predicted bacterial genes associated with level 1 KEGG pathways varies significantly across the macroscopic genome. (**B**) The abundance of level 2 KEGG pathways and functional pathways. To determine functional and phenotypic differences in *T. s. elegans* in different pH groups, bacterial phenotypes were analyzed and predicted using the BugBase algorithm. (**C**) Based on one-way ANOVA of phenotypic differences between groups, the vertical axis represents the phenotype name, and the horizontal axis represents the percentage value of the relative abundance of a certain phenotype in the sample, with significant intergroup differences denoted by *. (**D**) Comparison of single phenotypes based on Kruskal–Wallis rank-sum tests and analysis of variance. The far right is the *p* value, * *p* < 0.05 and different colors represent different groups.

## Data Availability

The data presented in this study are deposited in the NCBI Sequence Read Archive (SRA) under accession number PRJNA 1102732.

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
