# Peer review of "Effects of Freshwater Acidification on the Gut Microbial Community of Trachemys scripta elegans"

_animals, 2024, doi:10.3390/ani14131898_

Round 1
Reviewer 1 Report
Comments and Suggestions for Authors
Freshwater acidification is a global environmental issue threatening freshwater ecosystems. This study investigated the changes in the gut microbiota of red-eared sliders (Trachemys scripta elegans) under acidic conditions. Significant differences were observed in β-diversity but not in α-diversity of the gut microbiota across different pH levels. The Gut Microbiome Health Index and Microbial Dysbiosis Index showed significant differences between more and less acidic environments. Firmicutes, Fusobacteria, and Bacteroidetes were the predominant phyla, with Fusobacteria showing notable variation. At the genus level, differences were found in Cetobacterium, Turicibacter, unclassified Eubacteriaceae, and Anaerorhabdus_furcosa_group. Functional predictions revealed changes in potentially pathogenic and stress-tolerant traits.
The study is well-designed and executed. However, it can further be improved by incorporating the following comments.
In Methods, mention how many T. s. elegans were in each group and how many samples were subjected to the rRNA gene-based sequencing.
There should be a uniform style; either use 16S rDNA or 16S rRNA gene
The discussion needs to be strengthened by incorporating and comparing the results of other similar studies
Why did the T. s. elegans fast for 3 days before sample collection, and how can it impact microbial diversity?
What are the limitations of this study i.e, it was not a natural environment, it was the laboratory-induced environment. small sample size
Comments on the Quality of English Languageminor correction needed
Reviewer 2 Report
Comments and Suggestions for Authors
The manuscript entitled"Effects of freshwater acidification on the gut microbial community of Trachemys scripta elegans" investigated the impact of freshwater acidification on the gut microbiota of the red-eared slider (Trachemys scripta elegans), revealing significant alterations in microbial composition and functionality in response to varying pH levels. It highlights the sensitivity of gut microbiota to environmental pH changes and the potential ecological implications for aquatic species' health and biodiversity. The findings are interesting and the manuscript is well-organized. However, there are few question should be addressed:
1. Line 10, "explores" should be "explored", some other mistakes like this should also be checked carefully.
2. Why did the author choose Ph leveles of 5.5 and 6.5 as the acidification condition in your experiment?
3. The histological results should also be presented, like intestinal or liver .
4. Recent related research should be introduced to enhance the disscussion.
Reviewer 3 Report
Comments and Suggestions for Authors
The paper entitle“Effects of freshwater acidification on the gut microbial community of Trachemys scripta elegans” used high-throughput Illumina sequencing, this research explores how freshwater acidification affects the gut microbiota of Trachemys scripta elegans. Authors revealed that shifts in pH cause notable changes in the microbes' composition and their ability to handle stress and disease. It is important to understand the key for managing freshwater ecosystems' ecological effects of acidification and for conservation efforts to protect their biodiversity. The paper is interesting and can provide more knowledge of gut microbial community. However, some problem need address before published the paper.
1. I think that authors should check the authors and number in page 1 line 3.
2. Introduction: authors read the references “Microbial communities play a crucial role in host health, as they are not only involved in regulating the host's immune system and promoting metabolism but also assist the host in resisting various environmental stresses”. I read the three paper and find that it is not related to microbial communities involved in regulating the host's immune system. Authors should read more references that microbial communities play a crucial role in host health.
3. Results: I do not understand why is this paragraph in bold. Please check it.
4. Discussion: The functional analysis of animal gut microbiota plays a crucial role in revealing the physiological functions and ecological adaptability of the host [20]. It is not right that the sentence only refers a published paper “Sepulveda, J.; Moeller, A.H. The effects of temperature on animal gut microbiomes. Front. Microbiol. 2020, 11. 489 ”. Also, it is not right. I think that there are so many new references that can reflect the physiological functions and ecological adaptability of the host. For instance, Sun et al., (2023) in Integrative Zoology reviews the paper “The honeybee gut resistome and its role in antibiotic resistance dissemination”. In addition, Chen et al., (2023). Altitude-dependent metabolite biomarkers reveal the mechanism of plateau pika adaptation to high altitudes. Integrative Zoology 18:1041-1055. Xu et al. (2023). Deterministic processes dominate microbial community assembly in artificially bred Schizothorax wangchiachii juveniles after being released into wild. Integrative Zoology18:1072-1088.
5. Discussion: I suggest that authors discuss significant differences in α-diversity among the gut microbial communities in the which should refer species of the same genus.
Round 2
Reviewer 3 Report
Comments and Suggestions for Authors
I am happy that authors have revised the paper based on my suggestions. I am looking forward to see the publishing paper.
Best regards
Wenbo Liao